# Associations of Parameters of the Tryptophan–Kynurenine Pathway with Cardiovascular Risk Factors in Hypertensive Patients

**DOI:** 10.3390/nu15020256

**Published:** 2023-01-04

**Authors:** Verena Theiler-Schwetz, Christian Trummer, Martin R. Grübler, Martin H. Keppel, Armin Zittermann, Andreas Tomaschitz, Winfried März, Andreas Meinitzer, Stefan Pilz

**Affiliations:** 1Division of Endocrinology and Diabetology, Department of Internal Medicine, Medical University of Graz, 8036 Graz, Austria; 2Regional Hospital Wiener Neustadt, 2700 Wiener Neustadt, Austria; 3Department of Aging Medicine and Aging Research, University of Zurich, 8006 Zurich, Switzerland; 4Department of Laboratory Medicine, Paracelsus Medical University Salzburg, 5020 Salzburg, Austria; 5Clinic for Thoracic and Cardiovascular Surgery, Herz-und Diabeteszentrum Nordrhein-Westfalen (NRW), Ruhr University Bochum, 32545 Bad Oeynhausen, Germany; 6Health Center Trofaiach-Gössgrabenstrasse, 8793 Trofaiach, Austria; 7SYNLAB Academy, Synlab Holding Deutschland GmbH, 68159 Mannheim, Germany; 8Vth Department of Medicine (Nephrology, Hypertensiology, Rheumatology, Endocrinology, Diabetology, Lipidology), Medical Faculty Mannheim, University of Heidelberg, 68167 Mannheim, Germany; 9Clinical Institute of Medical and Chemical Laboratory Diagnostics, Medical University of Graz, 8036 Graz, Austria

**Keywords:** tryptophan, kynurenine, kynurenic acid, cardiovascular risk factors, arterial hypertension

## Abstract

Accumulating evidence suggests an association of the tryptophan–kynurenine (TRP-KYN) pathway with atherosclerosis and cardiovascular risk factors. In this cross-sectional analysis we investigated whether TRP-KYN pathway parameters are associated with 24 h blood pressure (BP) and other risk factors in patients with arterial hypertension from a tertiary care centre. In 490 participants, we found no significant and independent association of 24 h systolic and diastolic BP with parameters of the TRP-KYN pathway. However, linear regression analyses of HDL as dependent and TRP, KYN and quinolinic acid (QUIN) as explanatory variables adjusted for BMI and sex showed significant associations. These were found for KYN, BMI and sex (unstandardised beta coefficient −0.182, standard error 0.052, *p* < 0.001; −0.313 (0.078), *p* < 0.001; −0.180 (0.024), *p* < 0.001, respectively) as well as for QUIN, BMI and sex (−0.157 (0.038), *p* < 0.001; −0.321 (0.079), *p* < 0.001; −0.193 (0.024), *p* < 0.001, respectively). Smokers had significantly lower levels of KYN (2.36 µmol/L, IQR 2.01–2.98, versus 2.71 µmol/L, IQR 2.31–3.27, *p* < 0.001), QUIN (384 nmol/L, IQR 303–448, versus 451 nmol/L, IQR 369–575, *p* < 0.001) and KYN/TRP ratio (38.2, IQR 33.7–43.2, versus 43.1, IQR 37.5–50.9, *p* < 0.001) compared to non-smokers. We demonstrated that TRP/KYN pathway metabolites are associated with some cardiovascular risk factors, warranting further studies to elucidate the diagnostic and therapeutic potential of the TRP-KYN pathway for cardiovascular diseases.

## 1. Introduction

Cardiovascular diseases (CVDs) are the leading cause of death worldwide [1] as the major known risk factors are related to lifestyle and the individual genetic background. Obesity, diabetes mellitus, hypercholesterolaemia, hypertension, smoking or age contribute to the development of atherosclerosis, a chronic inflammatory disease affecting large- and medium-sized arteries, which is, by itself, the most important single cause of CVD. Recent evidence suggests a potential role of metabolites of the essential amino acid tryptophan (TRP) in the initiation and progression of atherosclerosis [2,3].

TRP is an essential amino acid required by all forms of life as a building block for protein synthesis [4] and is obtained mostly from protein-based foods such as milk, soy products and egg white [5,6,7]. It is metabolised via the serotonin/melatonin, the kynurenine (KYN) and the indole pathways. The latter involves the direct transformation of TRP by intestinal microorganisms into indole and its derivatives [8].

The KYN pathway accounts for more than 95% of peripheral catabolism of free TRP and leads to the production of several biologically active metabolites [3,9,10], which seem to play a fundamental immunomodulatory role [10]. KYN is the central intermediate of the pathway and is metabolised further by three enzymes. The end products of these three pathways are (a) xanthurenic acid, (b) quinolinic acid (QUIN) and picolinic acid as well as (c) kynurenic acid (KA). The enzyme indoleamine 2,3-dioxygenase (IDO) catalyses the first and rate limiting step of the TRP metabolism along the KYN pathway. IDO has two isoforms, IDO1 and IDO2, IDO1 being the major one controlling tryptophan degradation [11].

Increasing evidence suggests an association between IDO activity and systemic chronic low-grade immune-mediated inflammation [12] in the context of the development of atherosclerosis and its sequelae [2]. In particular, the pro-inflammatory Th1-type cytokine interferon-γ (IFN-γ) activates IDO in monocyte-derived macrophages, dendritic cells and other cells. Under these inflammatory conditions, the upregulation of IDO causes an increased degradation of TRP to KYN [10]. Patients with CVD often have increased serum KYN/TRP ratios [2] which seem to correlate with the severity of atherosclerosis and might play a potential role in predicting CVD [10,13].

Parameters of the TRP-KYN pathway have also been associated with cardiovascular risk factors per se, such as arterial hypertension, obesity and diabetes mellitus [3,14,15,16,17,18], with arterial hypertension being the most important cardiovascular risk factor when considering the global burden of disease [19]. Whether parameters of the TRP-KYN pathway are associated with 24 h blood pressure (BP) has not been investigated to date. 

We therefore aim to investigate whether parameters of the KYN-TRP pathway are associated with 24 h systolic and diastolic BP as well as other cardiovascular risk factors in patients with arterial hypertension.

## 2. Materials and Methods

### 2.1. Study Design

In this cross-sectional investigation, we included 490 study participants who were screened for inclusion in a vitamin D randomized, controlled trial (RCT) [20] performed at the Medical University of Graz, Austria. Only the baseline laboratory and 24 h ambulatory BP monitoring (ABPM) measurements of the original trial were for used for the present analysis. The trial was registered at EU Clinical Trials Register (http://www.clinicaltrialsregister.eu, accessed 16 February 2011, EudraCT number 2009-018125-70) and clinicaltrials.gov, accessed on 25 December 2022 (ClinicalTrials.gov Identifier NCT02136771). All publications of this trial adhere to the Consolidated Standards of Reporting Trials (CONSORT) 2010 statement [21].

### 2.2. Study Participants

The study participants of the Styrian Vitamin D Hypertension Trial were 18 years or older and had arterial hypertension. Arterial hypertension was diagnosed based on valid guidelines at the time when patients were included [22], i.e., an office systolic BP of ≥140 mmHg or diastolic BP ≥ 90 mmHg, a mean systolic 24 h ambulatory ABPM of ≥125 mmHg or diastolic ≥ 80 mmHg, a systolic home BP of ≥130 mmHg or diastolic ≥ 85 mmHg or antihypertensive medication. As published before [20], exclusion criteria for the RCT were 25-hydroxyvitamin D (25(OH)D) levels above 30 ng/mL, high levels of calcium, acute coronary syndrome or cerebrovascular events 2 weeks prior to screening, pregnancy or current breast feeding, drug intake due to participation in another clinical study, an estimated glomerular filtration rate < 15 mL/min per 1.73 m^2^, recent changes in antihypertensive medication in the 4 weeks prior to screening or intended change of antihypertensive treatment, diagnosed comorbidity with an estimated life expectancy shorter than 12 months, 24 h systolic blood pressure > 160 mmHg or <120 mmHg, 24 h diastolic blood pressure > 100 mmHg, acute diseases requiring relevant drug therapy, chemotherapy or radiation or a regular daily intake of more than 880 IU of vitamin D in the 4 weeks prior to screening. While these above described criteria apply for inclusion/exclusion in the RCT, we included all study participants in our present investigation who were screened and had available data for the 24 h ABPM and parameters of the KYN-TRP pathway regardless of their 25-hydroxvitamin D levels.

All study participants provided written informed consent before study inclusion. The ethics committee of the Medical University of Graz, Austria, approved the study, which was designed in accordance with the Declaration of Helsinki. Participants were recruited from the outpatient clinics of the Division of Cardiology and the Division of Endocrinology and Diabetology, Department of Internal Medicine, Medical University of Graz, Graz, Austria, throughout the year from June 2011 until August 2014.

### 2.3. Measurements

Study visits with patient interviews, physical examinations and blood samplings were performed between 7 a.m. and 11 a.m. after an overnight fast, and detailed methods of this trial have been published previously [20]. On the day of the first visit, ABPM measurements and 24 h urine collections were started. Patients returned to the outpatient clinic the following day to return the ABPM equipment as well as the collected urine.

We employed a validated 24 h ABPM device (Spacelabs 90217A; Spacelabs Healthcare, Inc., Issaquah, WA, USA) to perform 24 h systolic and diastolic BP measurements, following the recommendations of the European Society of Hypertension [23]. The appropriate cuff for BP recordings was chosen according to the circumference of the patient’s upper arm. BP measurements were repeated in 15 min intervals during the day (6 a.m. to 10 p.m.) and every 30 min during the night (10 p.m. to 6 a.m.). The Spacelabs 90217 device complies with the standard of the Association for the Advancement of Medical Instrumentation and has received the highest British Hypertension Society grade of ‘A’ for systolic and for diastolic blood pressures [24].

Laboratory parameters were measured by routine laboratory procedures on a daily basis. For the measurement of parameters of the TRP-KYN pathway, serum was centrifuged, aliquoted and frozen at −80 °C until KYN, TRP and KA were measured in serum using high-performance liquid chromatography with a simultaneous ultraviolet and fluorimetric detection system [25]. Within-day coefficients of variation (CVs) at different concentrations were between 1.7% to 4.3% for KYN, 0.7% to 2.9% for TRP and and 2.6 to 4.5% for KA. Between-day CVs were in the range of 2.0% to 5.4% and 6.3% to 9.3% and 8.4% to 11.6%, respectively. QUIN was determined in serum using a validated liquid chromatography tandem mass spectrometry method [26]. Within-day CVs were 4.5% (225 nmol/L) and 1.2% (725 nmol/L), and between-day CVs 7.2% (235 nmol/L) and 6.3% (752 nmol/L).

### 2.4. Statistical Methods

Continuous data with a normal distribution as determined by Kolmogorov–Smirnov tests and data visualisation by histogram are reported as means with standard deviations. Non-normally distributed variables are shown as medians with interquartile ranges (IQR), while categorical data are presented as percentages. Group comparisons at baseline were analysed by the application of an unpaired Student’s t-test, chi-squared test or Mann–Whitney U test. Where appropriate, skewed variables were log(e) transformed to apply parametric statistical analyses. Analyses of variance (ANOVA) were used to evaluate associations between measures of the TRP-KYN pathway grouped into quartiles and cardiovascular risk factors as outcome variables. For parameters with significant associations in ANOVA, we used linear regression analyses with measures of the TRP-KYN pathway as explanatory variables and cardiovascular risk factors as dependent variables with adjustment for age, sex, CRP and BMI (if applicable).

Bonferroni correction for *p*-values was applied to account for multiple testing (*p* = 0.05/40). Thus, *p*-values < 0.00125 were considered statistically significant. Statistical analyses were performed with SPSS version 27 (SPSS, Chicago, IL, USA).

## 3. Results

In total, 518 hypertensive patients were assessed for eligibility in the Styrian Vitamin D Hypertension Trial, and 490 patients with available 24 h systolic and diastolic blood pressure and parameters of the KYN-TRP pathway were suitable for the present analysis. Baseline characteristics for the whole study cohort and for women and men separately are provided in Table 1, haemodynamic and metabolic parameters are shown in Table 2, parameters of the TRP-KYN pathway are shown in Table 3. There was a statistically significant correlation between age and TRP (Spearman beta coefficient −0.193, *p* < 0.001), KYN (Spearman beta coefficient 0.288, *p* < 0.001), KA (Spearman beta coefficient 0.148, *p* = 0.001), QUIN (Spearman beta coefficient 0.390, *p* < 0.001) and KYN/TRP ratio (Spearman beta coefficient 0.418, *p* < 0.001).

Table 4 shows the associations between quartiles of TRP, KYN, KA, QUIN and KYN/TRP ratio with cardiovascular risk factors. There was no statistically significant difference in 24 h systolic BP across quartiles for any of the parameters studied. There was, however, a significantly lower 24 h diastolic BP across quartiles of KYN/TRP ratio comparing the highest with the lowest quartile (*p* < 0.001). Linear regression analyses of 24 h diastolic BP as the dependent variable and KYN/TRP ratio as explanatory variables with adjustment for various covariates (age, sex, BMI and CRP) showed that the association with the KYN/TRP ratio was no longer significant (unstandardised beta coefficient −0.215, standard error 0.026, *p* < 0.001 for age and unstandardised beta coefficient 0.046, standard error 0.010, *p* < 0.001, for sex).

For lipid parameters, linear regression analyses of LDL as the dependent variable and KYN as the explanatory variable with adjustments for various covariates (age, sex, BMI and CRP) showed no significant association with KYN or any of the covariates. Linear regression analyses of HDL as the dependent variable and TRP as the explanatory variable with adjustments for the same covariates, showed significant associations only for BMI and gender, but not for TRP. Linear regression analyses of HDL as the dependent variable and KYN as the explanatory variable with adjustments for covariates showed significant associations for KYN, BMI and sex (unstandardised beta coefficient −0.182 (standard error 0.052), *p* < 0.001, for KYN, −0.313 (0.078), *p* < 0.001, for BMI, −0.180 (0.024), *p* < 0.001, for sex). Linear regression analyses of HDL as the dependent variable and QUIN as the explanatory variable with adjustment for covariates showed significant associations for QUIN (unstandardised beta coefficient −0.157 (standard error 0.038), *p* < 0.001), BMI (−0.321 (0.079), *p* < 0.001) and sex (−0.193 (0.024), *p* < 0.001).

Regarding diabetes mellitus, we observed an increase in diabetes prevalence across KYN and KYN/TRP ratio quartiles, losing significance after correction for multiple testing (Table 2). Compared to hypertensive patients without a diagnosis of diabetes, hypertensive patients with type 2 diabetes had elevated levels of KYN (2.59 μmol/L, IQR 2.20–3.06, without diabetes vs. 2.87 μmol/L, IQR 2.38–3.55, with diabetes, *p* < 0.001). The difference in KYN/TRP ratios (41.4, IQR 36.1–48.0, without diabetes vs. 44.6, IQR 37.0–54.0, with diabetes, *p* = 0.009), KA (39.1 nmol/L, IQR 31.3–51.0, vs. 43.5 nmol/L, IQR 34.2–53.4, *p* = 0.026) and QUIN (428 nmol/L, IQR 362–534 vs. 471 nmol/L, IQR 352–616, *p* = 0.028) did not remain significant after adjustment for multiple testing. There was no difference in TRP (63.1 μmol/L, IQR 56.8–68.9 vs. 62.2 μmol/L, IQR 56.7–69.5, *p* = 0.791) between the groups.

There was a decrease in smoking prevalence across KYN quartiles, QUIN quartiles and KYN/TRP ratio quartiles (Table 2). Active smokers, as compared to non-smokers, had lower levels of KYN (2.36 µmol/L, IQR 2.01–2.98, versus 2.71 µmol/L, IQR 2.31–3.27, *p* < 0.001), QUIN (384 nmol/L, IQR 303–448, versus 451 nmol/L, IQR 369–575, *p* < 0.001) and KYN/TRP ratio (38.2, IQR 33.7–43.2, versus 43.1, IQR 37.5–50.9, *p* < 0.001).

## 4. Discussion

We observed no significant and independent association between parameters of the TRP-KYN pathway and 24 h BP measurements. For other cardiovascular risk factors, we found statistically significant associations between parameters of the TRP-KYN pathway with BMI, age, sex, HDL, smoking and diabetes mellitus.

Evidence from mouse studies has previously suggested that parameters of the TRP-KYN pathway are associated with arterial hypertension [14,15]. Experimentally induced endothelial expression of IDO in mice resulted in decreased plasma TRP and increased KYN and hypotension. KYN dose-dependently lowered blood pressure in spontaneously hypertensive rats, inhibited contraction of arteries, and relaxed pre-constricted rings endothelium-independently [14]. The vasodilatory effects of KYN were attributed to the activation of guanylate cyclase and adenylate cyclase [14]. Clinically, it could be demonstrated that TRP degradation to KYN showed a correlation with hypotension during sepsis in humans [27]. Clinical studies showed higher KYN levels in hypertensive as compared to normotensive patients [15]. Our findings likewise support an association of KYN/TRP ratio, a proxy for IDO activation, with a decrease in blood pressure, even though our findings did not remain significant after adjustment for multiple confounders. The fact that we did not observe a significant association of parameters of the TRP-KYN pathway and 24 h systolic and diastolic blood pressure does not necessarily contradict previously found results showing alterations in TRP-KYN pathway in patients with hypertension. Reasons for divergent results could be the fact that all our patients included had arterial hypertension to begin with and many clinical studies have found differences in TRP-KYN parameters when comparing patients with hypertension to healthy controls. Further, most of our patients were on anti-hypertensive treatment, possibly attenuating potential associations.

The associations seen in our patients between parameters of TRP-KYN pathway and BMI have been well documented before. In accordance with our results, obese patients have previously been shown to have higher levels of KYN and KYN/TRP ratios as compared to lean patients [2,28,29]. In addition, IDO expression in omental and subcutaneous adipose tissue and in the liver has been found increased and inversely correlated with arterial blood pressure in obese subjects [28]. The increased TRP degradation in both fat tissues and the liver support a role of this metabolic pathway in coronary artery disease and other obesity-associated pathologies [28].

In line with this, KYN was also higher in our hypertensive patients with diabetes as compared to those without a diagnosis of diabetes, in support of previously published data [15,30]. Matching these findings, TRP and KYN have also been associated with and have improved the prediction of all-cause mortality in type 2 diabetes in 856 individuals with 279 all-cause deaths [31].

Data on lipid metabolism and TRP-KYN pathway are scarce; those available support our findings of a negative association of KYN with HDL by showing a negative correlation of IDO activity with HDL in young Finnish adults as well as in an older Finnish population [32,33]. Administration of the TRP metabolite 3-hydroxyanthranilic acid to LDL receptor knock-out mice significantly decreased total plasma cholesterol and triglyceride levels and significantly increased HDL cholesterol [34].

Based on current data, however, we are still lacking a complete understanding of the role of the TRP-KYN pathway in the modulation of energy and lipid homeostasis. It is currently still unclear whether elevated plasma levels reflect a causal pathway or represent only epiphenomena of disease development. If the first were to hold true, targeting the KYN pathway might harbour potential to expand the range of therapies to prevent and treat metabolic diseases [35,36].

In addition to metabolic diseases and blood pressure, we were also able to show an association with parameters of the TRP-KYN pathway and smoking. While some groups failed to show such associations [37], data pointing towards lower KYN/TRP ratios reflecting IDO activity and lower KYN levels in smokers as compared to non-smokers have been shown [15,38,39]. Smoking was also associated with a lower pyridoxal 5′-phosphate (PLP) status [37,40], which is a cofactor for kynurenine aminotransferases mediating the irreversible and permanent transamination of KYN to KA. Decreased serum IDO activity suggests that components of cigarette smoke may potentially exert immunosuppressive effects upon serum IDO activity and the reduction in IDO-dependent immunosuppression could thus be responsible for the known immunostimulatory effects of smoking [38].

To underline the validity of our data, we have shown significant associations between age and all parameters of the TRP-KYN pathway. These findings, presumably attributed to the upregulation of the TRP-KYN pathway in ageing due to the activation of IDO by age-related chronic inflammation [41], are well known from animal studies [42] as well as clinical data [15,43,44]. Further, our results confirm previously published literature showing lower levels of parameters of the TRP-KYN pathway in women [44]. We also have to acknowledge that our data are limited due to the nature of our cross-sectional study design so that we cannot exclude residual confounding. Additionally, due to the lack of a control group, we cannot generalise our findings to healthy individuals. Finally, our cohort of hypertensive patients referred to a tertiary care centre limits the generalisability of our findings and may have introduced some sort of selection bias. The main strength of our work is the well characterised cohort and that we were the first to evaluate associations between the TRP-KYN pathway and 24 h BP measurements.

In conclusion: we did not show a significant association of TRP-KYN pathway with 24 h blood pressure in hypertensive patients. Our data do, however, emphasise a significant negative association of KYN and HDL as well as an association of KYN and IDO activity with smoking status. Our results suggest that the TRP-KYN pathway might be involved in regulation of cardiovascular risk factors and might constitute a novel marker for risk stratification and determination of cardiovascular disease prognosis. Whether the TRP-KYN pathway may also be a promising drug target warrants further in-depth investigations.

## Figures and Tables

**Table 1 nutrients-15-00256-t001:** Selected baseline characteristics of all study participants.

Variables	All Study Participants	Women	Men
Numbers	490	258	232
Age (years)	62.8 (55.3–68.7)	63.0 (56.3–68.3)	62.6 (53.4–69.1)
Body mass index (kg/m^2^)	28.7 (26.1–32.3)	28.4 (25.9–32.7)	29.1 (26.3–31.9)
Active smoker (%)	14.9	14.8	15.1
Previous myocardial infarction (%)	6.2	10.8	8.4
Antihypertensive drugs (n)	2 (1–3)	2 (1–3)	2 (1–3)
ACE-inhibitor (%)	34.6	33.1	36.2
AT II blocker (%)	31.7	31.9	31.5
Thiazide diuretic (%)	40.5	41.6	39.2
Beta blocker (%)	49.7	49.4	50.0
Calcium channel blocker (%)	26.4	24.1	28.9

Data are presented as medians with interquartile ranges or percentages. ACE-inhibitor = angiotensin-converting-enzyme inhibitors, AT II blocker = angiotensin II receptor blockers.

**Table 2 nutrients-15-00256-t002:** Haemodynamic and metabolic parameters of all study participants.

Variables	All Study Participants	Women	Men
Office systolic blood pressure (mmHg)	141.0 (129.0–151.0)	139.0 (126.5–150.5)	141.0 (130.0–152.5)
Office diastolic blood pressure (mmHg)	85.0 (79.0–92.0)	84.0 (78.0–90.0)	86.5 (80.0–95.0)
24 h systolic blood pressure (mmHg)	126.1 (118.6–135.7)	123.2 (116.0–131.4)	128.9 (121.6–137.9)
24 h diastolic blood pressure (mmHg)	75.1 (70.1–82.1)	73.8 (68.8–79.4)	76.4 (72.7–84.3)
Resting heart rate (beats/minute)	61 (55–69)	61 (56–69)	61 (55–69)
25-hydroxyvitamin D (ng/mL)	26.7 (20.2–35.5)	27.0 (20.1–35.3)	26.6 (20.3–35.7)
eGFR (mL/min/1.73 m^2^)	79.6 ± 17.6	78.3 ± 17.7	81.2 ± 17.4
Diabetes mellitus (%)	23	32.8	27.6
Fasting glucose (mg/dL)	97.0 (89.0–115.0)	95.0 (88.0–108.0)	99.5 (90.3–131.8)
HbA1c (mmol/mol)	5.7 (5.4–6.3)	5.7 (5.5–6.1)	5.8 (5.4–6.8)
HOMA-IR	1.66 (1.01–3.03)	1.61 (0.98–2.75)	1.80 (1.03–3.54)
Triglycerides (mg/dL)	110 (75–152)	99 (73–141)	116 (77–167)
Total cholesterol (mg/dL)	197 (166–225)	206 (176–233)	186 (162–215)
HDL-cholesterol (mg/dL)	56 (46–69)	62 (52–75)	51 (43–62)
LDL-cholesterol (mg/dL)	112 (88–142)	119 (90–146)	104 (87–136)
PWV (m/sec)	8.2 (7.1–9.5)	7.9 (6.8–9.2)	8.6 (7.5–9.8)
CRP (mg/L)	1.8 (0.8–3.4)	2.0 (0.9–3.7)	1.7 (0.8–3.2)

Data are presented as means with standard deviations, medians with interquartile ranges or percentages. eGFR = estimated glomerular filtration rate, HbA1c = glycated haemoglobin, HOMA-IR = homeostatic model assessment of insulin resistance, HDL = high-density lipoprotein, LDL = low-density lipoprtoein, PWV = pulse wave velocity, CRP = C-reactive protein.

**Table 3 nutrients-15-00256-t003:** Metabolites of the tryptophan–kynurenine pathway of all study participants.

Variables	All Study Participants	Women	Men
Kynurenine (µmol/L)	2.98 (2.63–3.54)	2.97 (2.64–3.54)	3.02 (2.62–3.57)
Kynurenic acid (nmol/L)	41.71 (34.57–54.01)	39.50 (33.40–50.98)	46.80 (35.99–56.94)
Tryptophan (µmol/L)	63.61 ± 10.14	61.7 ± 9.7	65.4 ± 10.3
Quinolin Acid (nmol/L)	434.15 (360.55–565.65)	431.75 (369.33–562.30)	437.25 (351.85–574.20)
Kynurenine/tryptophan ratio	42.2 (36.4–49.5)	42.5 (36.9–50.8)	41.9 (35.9–49.1)

Data are presented as means with standard deviations or medians with interquartile ranges.

**Table 4 nutrients-15-00256-t004:** Associations of tryptophan, kynurenine, quinolinic acid, kynurenic acid and kynurenine/tryptophan ratio with cardiovascular risk factors.

	Quartiles				*p*-Value
	**Tryptophan quartiles**				
	**1st quartile**	**2nd quartile**	**3rd quartile**	**4th quartile**	
**Number**	120	121	123	125	
Tryptophan (µmol/L)	<56.8	56.8–62.9	63.0–68.8	>68.8	
	52.0 (47.9–54.4)	59.2 (57.8–60.7)	65.2 (64.1–66.9)	74.0 (71.1–78.2)	
24 h systolic BP (mmHg)	128.0 (118.7–134.1)	123.4 (116.1–134.9)	125.8 (117.2–134.4)	128.1 (121.5–136.8)	0.074
24 h diastolic BP (mmHg)	74.4 (70.7–80.9)	74.9 (69.3–79.1)	74.9 (69.1–81.8)	78.8 (72.8–85.1)	**0.005**
BMI (kg/m^2^)	27.7 (25.4–30.6)	30.0 (26.3–34.0)	28.4 (26.2–31.4)	29.3 (26.8–32.4)	**0.014**
Total cholesterol (mg/dL)	194 (169–228)	198 (169–230)	198 (162–222)	197 (164–221)	0.832
LDL cholesterol (mg/dL)	109 (89–145)	114 (88–142)	113 (83–143)	112 (89–137)	0.897
HDL cholesterol (mg/dL)	59 (52–76)	61 (48–70)	55 (45–68)	53 (43–64)	**0.001**
Diabetes mellitus (%)	28.3	28.3	25.2	28.8	0.917
Active smoker (%)	11.7	10	17.9	20	0.083
	**Kynurenine quartiles**				
	**1st quartile**	**2nd quartile**	**3rd quartile**	**4th quartile**	
Number	122	123	121	124	
Kynurenine (μmol/L)	<2.24	2.24–2.62	2.63–3.17	>3.17	
	2.00 (1.85–2.13)	2.44 (2.35–2.53)	2.88 (2.77–3.04)	3.64 (3.42–3.92)	
24 h systolic BP	124.8 (117.3–132.3)	126.7 (119.0–136.0)	124.4 (117.3–136.4)	128.1 (120.8–136.7)	0.447
24 h diastolic BP	76.0 (72.1–83.2)	76.6 (69.9–83.0)	74.9 (70.2–79.6)	74.2 (68.2–81.1)	0.074
BMI (kg/m^2^)	27.0 (24.5–31.2)	28.3 (25.9–30.7)	28.9 (26.7–32.9)	30.4 (27.6–34.2)	**<0.001**
Total cholesterol (mg/dL)	204 (171–230)	206 (177–234)	191 (164–223)	183 (152–218)	0.052
LDL cholesterol	115 (94–147)	125 (98–145)	110 (84–142)	101 (80–134)	**0.001**
HDL cholesterol (mg/dL)	61 (49–76)	59 (48–72)	54 (44–67)	53 (43–62)	**<0.001**
Diabetes mellitus (%)	21.3	23	26.4	39.5	**0.006**
Active smoker (%)	27	10.7	15.7	6.5	**<0.001**
	**Quinolinic acid**				
	**1st quartile**	**2nd quartile**	**3rd quartile**	**4th quartile**	
Number	114	114	115	115	
Quinolinic acid (nmol/L)	<361	361–433	434–565	>565	
	316 (287–340)	402 (379–417)	494 (459–521)	669 (603–802)	
24 h systolic BP	124.9 (116.9–133.9)	124.3 (118.5–131.0)	127.9 (118.1–136.8)	125.7 (118.8–136.8)	0.71
24 h diastolic BP	77.3 (72.0–83.9)	75.5 (71.7–81.5)	73.9 (69.0–80.9)	73.5 (68.0–80.7)	**0.004**
BMI (kg/m^2^)	27.7 (24.9–30.9)	27.6 (25.7–30.6)	29.4 (26.6–32.7)	30.6 (27.5–34.2)	**<0.001**
Total cholesterol (mg/dL)	199 (169–229)	206 (178–231)	198 (164–223)	185 (155–219)	0.041
LDL cholesterol (mg/dL)	113 (89–141)	119 (97–148)	121 (89–142)	107 (82–138)	0.064
HDL cholesterol (mg/dL)	58 (50–75)	60 (48–74)	56 (45–68)	51 (40–62)	**<0.001**
Diabetes mellitus (%)	26.5	14.9	21.7	36.5	**0.002**
Active smoker (%)	24.8	17.5	9.6	5.2	**<0.001**
	**Kynurenic acid**				
	**1st quartile**	**2nd quartile**	**3rd quartile**	**4th quartile**	
Number	122	123	122	123	
Kynurenic acid (nmol/L)	<32.0	32.0–40.0	40.1–52.2	>52.2	
	26.2 (22.6–29.5)	36.2 (34.2–38.3)	46.4 (42.8–49.1)	61.8 (55.4–72.0)	
24 h systolic BP	123.1 (117.6–132.0)	126.4 (118.0–135.2)	125.4 (117.9–134.1)	128.8 (120.7–139.9)	0.104
24 h diastolic BP	75.5 (70.0–80.5)	75.0 (70.1–83.0)	74.9 (69.0–81.8)	75.1 (70.7–83.0)	0.695
BMI (kg/m^2^)	27.0 (24.4–29.8)	29.4 (25.4–32.6)	28.7 (26.4–31.7)	30.3 (27.6–35.2)	**<0.001**
Total cholesterol (mg/dL)	192 (165–220)	199 (171–225)	205 (166–235)	192 (159–221)	0.105
LDL cholesterol (mg/dL)	108 (89–143)	116 (91–139)	118 (88–150)	108 (87–137)	0.057
HDL cholesterol (mg/dL)	58 (48–71)	56 (47–72)	57 (47–71)	54 (44–63)	0.02
Diabetes mellitus (%)	21.3	25.4	31.1	32.5	0.174
Active smoker (%)	23	17.2	8.2	11.4	**0.007**
	**Kynurenine/tryptophan ratio**				
	**1st quartile**	**2nd quartile**	**3rd quartile**	**4th quartile**	
Number	122	122	123	123	
Kynurenine/tryptophan ratio	<36.4	36.4–42.1	42.2–49.5	>49.5	
	32.6 (30.5–34.8)	39.8 (38.2–41.0)	45.6 (44.1–47.4)	56.9 (52.8–66.3)	
24 h systolic BP	124.5 (117.5–132.9)	127.2 (119.7–138.0)	126.3 (118.7–134.1)	126.7 (118.3–136.5)	0.575
24 h diastolic BP	77.1 (72.8–83.3)	75.7 (70.8–83.6)	75.6 (70.1–80.9)	72.9 (67.9–78.9)	**<0.001**
BMI (kg/m^2^)	27.0 (24.5–30.5)	29.3 (26.3–32.4)	28.5 (26.6–31.3)	30.0 (26.8–34.7)	**<0.001**
Total cholesterol (mg/dL)	202 (171–224)	207 (172–232)	190 (169–227)	183 (152–221)	**0.002**
LDL cholesterol (mg/dL)	115 (94–144)	125 (97–145)	104 (88–140)	106 (75–137)	**0.003**
HDL cholesterol (mg/dL)	60 (47–72)	57 (48–73)	56 (48–69)	55 (43–63)	**0.002**
Diabetes mellitus (%)	23.8	23.1	23.6	39.8	**0.006**
Active smoker (%)	27.9	14	13	4.9	**<0.001**

Data are shown as medians with SD or as medians with interquartile ranges. Analysis of variance (ANOVA) was calculated. Non-parametric data were log(e) transformed for ANOVA, but untransformed values are shown. *p*-values in bold indicate statistically significant results.

## Data Availability

Not applicable.

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
