# Peer review of "Associations of Parameters of the Tryptophan–Kynurenine Pathway with Cardiovascular Risk Factors in Hypertensive Patients"

_nutrients, 2023, doi:10.3390/nu15020256_

Round 1

Reviewer 1 Report

In presented article entitled “Associations of parameters of the tryptophan-kynurenine pathway with cardiovascular risk factors in hypertensive patients authors evaluated correlations between metabolites of tryptophan-kynurenine pathway and cardiovascular disease risk factors in patients with hypertension. Scientific topic is interesting and worth investigating, however; there are several methodological errors and aspects that will be addressed below.

Major concerns

1.      Scientific value of presented data is limited due to the fact that no comparison with healthy controls was performed. Additionally, patients in this project characterize very narrow and limited representation. Based on this it is difficult to translate these results to more objective conclusions. Authors included above mentioned limitations in discussion what is appreciated, but still these aspects significantly affect value of presented project.

2.      In introduction authors wrote that there are 2 pathways of tryptophan metabolism ruling out synthesis of bacterial metabolites (indoles) of tryptophan. This 3rf pathways gains  scientific interest showing new promising effects of tryptophan metabolism. This aspect should be included in introduction.

3.      Authors wrote in Materials and Methods that: “As has been previously published [15], exclusion criteria for the RCT included a 25-hydroxyvitamin D (25(OH)D) concentration above 30 ng/mL”  This fact is important and was not justified by the authors. Proper synthesis of vitamin D has many beneficial effects in mammals. Sufficient synthesis of vit D is associated with levels of 25(OH) above 30 ng/mL. Since authors chose in this study participants with levels below this concentration reception of the whole article changes showing that in fact authors investigated metabolism of tryptophan in hypertensive patients with diagnosed vit D deficiency. Ruling out this aspect and lack of explanation is a serious flaw of presented article.

4.      Methodology section lack information on how many blood pressure measurements and sample collections were performed. What was the time of observation once it is “between June 2011 and August 2014”, and then it is “Study visits with patient interviews, physical examinations, and blood samplings 113 were performed throughout the year”.

5.      How it is possible that all groups had the same resting heart rate with such small range? Resting heart rate (beats/minute) 61 (55 69) 61 (56 69) 61 (55 69) they are almost identical. How was it calculated/measured?

6.      Presentation of data is far from ideal. 2 tables with all data at once diminish readability of the project. I would suggest dividing them into more tables, 1 with description of population, 2 with hemodynamic and metabolic data, 3 with metabolites of TRP and 4 as the current table 2. Additional figures would also made this article more interesting. With description of inclusion and exclusion criteria and how final population was established.

Minor concerns

1.      Please carefully check abbreviations within the text of article, especially when it comes to tryptophan-kynurenine pathway. Abbreviations should be explained when first introduced or separate list should be prepared. In text there several forms of “TRP-KYN pathway”, “KYN-TRP pathway”, “KP” – last not explained when introduced.

2.      2. Authors did not provide keywords in presented manuscript

Author Response

Comments and Suggestions for Authors

In presented article entitled “Associations of parameters of the tryptophan-kynurenine pathway with cardiovascular risk factors in hypertensive patients authors evaluated correlations between metabolites of tryptophan-kynurenine pathway and cardiovascular disease risk factors in patients with hypertension. Scientific topic is interesting and worth investigating, however; there are several methodological errors and aspects that will be addressed below.

Major concerns

1.      Scientific value of presented data is limited due to the fact that no comparison with healthy controls was performed. Additionally, patients in this project characterize very narrow and limited representation. Based on this it is difficult to translate these results to more objective conclusions. Authors included above mentioned limitations in discussion what is appreciated, but still these aspects significantly affect value of presented project.

We thank the reviewer for this comment. We agree that one major limitation is the lack of comparison to a healthy control group. However, we believe that the relatively large number of hypertensive patients investigated is a strength, together with the 24-hour ambulatory blood pressure measurements. We agree that generalisability may be limited, but since high blood pressure is the leading Global Burden of Disease risk factor worldwide accounting for millions of deaths and disability-adjusted life-years (GBD 2017 Risk Factor Collaborators, Lancet 2018, DOI: 10.1016/S0140-6736(18)32225-6), we may at least attempt to draw cautious conclusions from our group of hypertensive women and men to other patients with arterial hypertension. Moreover, the characteristics of our cohort cover a wide distribution regarding age and common cardiovascular risk factors. In the Discussion section of the manuscript, we added the following sentence:

“Also, due to the lack of a control group, we cannot generalise our findings to healthy individuals."

2.      In introduction authors wrote that there are 2 pathways of tryptophan metabolism ruling out synthesis of bacterial metabolites (indoles) of tryptophan. This 3rf pathways gains  scientific interest showing new promising effects of tryptophan metabolism. This aspect should be included in introduction.

We are grateful for this comment and added this information to the Introduction section of the manuscript, which now reads:

“It is metabolised via the serotonin/melatonin, the kynurenine (KYN) and the indole pathway. The latter involves the direct transformation of TRP by intestinal microorganisms into indole and its derivatives {{1789 Agus, Allison 2018;}}.”

3.      Authors wrote in Materials and Methods that: “As has been previously published [15], exclusion criteria for the RCT included a 25-hydroxyvitamin D (25(OH)D) concentration above 30 ng/mL”  This fact is important and was not justified by the authors. Proper synthesis of vitamin D has many beneficial effects in mammals. Sufficient synthesis of vit D is associated with levels of 25(OH) above 30 ng/mL. Since authors chose in this study participants with levels below this concentration reception of the whole article changes showing that in fact authors investigated metabolism of tryptophan in hypertensive patients with diagnosed vit D deficiency. Ruling out this aspect and lack of explanation is a serious flaw of presented article.

We fully agree with the reviewer, but would like to clarify. In the Methods section, we are indeed stating the in- and exclusion criteria of the original randomised controlled trial. As specified below, however, in the present analysis we have included all participants screened with available 24-hour blood pressure measurements, regardless of their vitamin D status.

The following statement is included in the manuscript.:

“While these above described criteria apply for inclusion/exclusion in the RCT, we included all study participants in our present investigation who were screened and had available data for the 24-hour BP measurement and parameters of the KYN-TRP pathway, regardless of their 25-hydroxvitamin D levels.”

To underline this and to avoid confusion regarding this issue, we have included data on 25-hydroxyvitamin D levels in the manuscript (please see Table 2) and hope that it is now clear for the reader. 

4.      Methodology section lack information on how many blood pressure measurements and sample collections were performed. What was the time of observation once it is “between June 2011 and August 2014”, and then it is “Study visits with patient interviews, physical examinations, and blood samplings 113 were performed throughout the year”.

Thank you for this comment, we would like to clarify. For the original study, blood sampling and 24-hour ambulatory blood pressure measurements were performed at baseline and after 8 weeks. For this analysis, however, we only used the baseline measurements. We added “throughout the year”, as this was relevant to the original study focussing on 25-hydroxyvitamin D levels. To clarify, we added the following sentence in the Methods section:

“Only the baseline laboratory and 24-hour ambulatory BP monitoring (ABPM) measurements of the original trial were for used for the present analysis.”

5.      How it is possible that all groups had the same resting heart rate with such small range? Resting heart rate (beats/minute) 61 (55 69) 61 (56 69) 61 (55 69) they are almost identical. How was it calculated/measured?

We have re-calculated the median heart rate with interquartile ranges for all participants and for men and women separately and can confirm the numbers are indeed correct. Heart rate in beats per minute was determined based on ECG, calculated from the 10s print in lead II multiplied by 6. 

6.      Presentation of data is far from ideal. 2 tables with all data at once diminish readability of the project. I would suggest dividing them into more tables, 1 with description of population, 2 with hemodynamic and metabolic data, 3 with metabolites of TRP and 4 as the current table 2. Additional figures would also made this article more interesting. With description of inclusion and exclusion criteria and how final population was established.

We have adapted the presentation of data as suggested by the reviewer to improve readability.  

To clarify which patients were included in this analysis, we tried to emphasise that all patients screened for the Styrian Vitamin D Hypertension Trial with available 24-hour ambulatory blood pressure monitoring and available parameters of the trypthophan-kynurenine pathway were included, regardless of their 25-hydroxyvitamin D levels (please see below).

“While these above described criteria apply for inclusion/exclusion in the RCT, we included all study participants in our present investigation who were screened and had available data for the 24-hour BP measurement and parameters of the KYN-TRP pathway regardless of their 25-hydroxvitamin D levels.”

Minor concerns

1.      Please carefully check abbreviations within the text of article, especially when it comes to tryptophan-kynurenine pathway. Abbreviations should be explained when first introduced or separate list should be prepared. In text there several forms of “TRP-KYN pathway”, “KYN-TRP pathway”, “KP” – last not explained when introduced.

Thank you for pointing this out, we have corrected this.

2.     Authors did not provide keywords in presented manuscript.

We have now provided keywords. 

Reviewer 2 Report

General comments

The report is based on a well-designed and conducted cross-sectional study performed in confirmed hypertensive patients. Although results pertaining to the association between the tryptophan-kynurenine pathway are mostly negative, they are important as they contribute to the understanding of etiology of CVD.  The adjusted association between high-density lipoprotein concentrations (dependent variable) and tryptophan, kynurenine and quinolinic acid (explanatory variables) provide some insight into the control of lipid metabolism. The use of the 24-hour blood pressure monitoring for this study is an asset. The authors should be commented for having performed an exhaustive statistical analysis. I have only few minor comments.

Specific comments

Please spell out the acronym KP (kynurenine pathway).

Could a mention on IDO isoforms be included. Indeed, IDO has two isoforms (IDO1 & IDO2), of which IDO1 is the major one controlling tryptophan metabolism (Ref.: Mengyu Li. Indoleamine 2,3-dioxygenase and ischemic heart disease: a Mendelian Randomization study. Sci Rep 2019;9:8491)? Could analysis with isoform as a variable provide more information?

References 12 & 13 (Line 69) do not address obesity and diabetes mellitus. Could more specific references be used to exemplify the point?

Author Response

General comments

The report is based on a well-designed and conducted cross-sectional study performed in confirmed hypertensive patients. Although results pertaining to the association between the tryptophan-kynurenine pathway are mostly negative, they are important as they contribute to the understanding of etiology of CVD.  The adjusted association between high-density lipoprotein concentrations (dependent variable) and tryptophan, kynurenine and quinolinic acid (explanatory variables) provide some insight into the control of lipid metabolism. The use of the 24-hour blood pressure monitoring for this study is an asset. The authors should be commented for having performed an exhaustive statistical analysis. I have only few minor comments.

Specific comments

Please spell out the acronym KP (kynurenine pathway).

We have corrected this. 

Could a mention on IDO isoforms be included. Indeed, IDO has two isoforms (IDO1 & IDO2), of which IDO1 is the major one controlling tryptophan metabolism (Ref.: Mengyu Li. Indoleamine 2,3-dioxygenase and ischemic heart disease: a Mendelian Randomization study. Sci Rep 2019;9:8491)? Could analysis with isoform as a variable provide more information?

We thank the reviewer for this comment and added information on the two IDO isoforms in the Introduction section. 

“IDO has two isoforms, IDO1 and IDO2, IDO1 being the major one controlling tryptophan degradation {{1790 Lob, Stefan 2009;}}.”

Due to the nature of our study, we were not able to analyse the expression of the tryptophan-kynurenine enzymes such as IDO1 in peripheral blood mononuclear cells. We do agree, of course, that this would be more accurate than determining the tryptophan-kynurenine ratio as a surrogate parameter. 

References 12 & 13 (Line 69) do not address obesity and diabetes mellitus. Could more specific references be used to exemplify the point?

We agree and have added more specific references. 

Round 2

Reviewer 1 Report

Authors in revised version of the manuscript included changes suggested by the reviewer. Overall reception of the article improved, however; some limitations still affect strength of presented results and their scientific value. 

I would advise including data on antyhypertensive treatment in Table 1 which consists of more demographic data. 

Authors emphasized limitations of the article in the discussion section.

There is a need for enabling reusability of once obtained data for sharing scientific results with others to extend knowledge on metabolites of tryptophan and their correlations with cardiovascular system. Presented results might bring scientific background for future research projects expanding knowledge on this interesting aspect of tryptophan metabolism. 

Taken together I am going to conditionally consider changes in the manuscript to be sufficient in authors' favour. 

Author Response

Rebuttal

Reviewer 1

Authors in revised version of the manuscript included changes suggested by the reviewer. Overall reception of the article improved, however; some limitations still affect strength of presented results and their scientific value. 

I would advise including data on antyhypertensive treatment in Table 1 which consists of more demographic data. 

We thank the reviewer for this comment and have changed the tables accordingly. 

Authors emphasized limitations of the article in the discussion section.

There is a need for enabling reusability of once obtained data for sharing scientific results with others to extend knowledge on metabolites of tryptophan and their correlations with cardiovascular system. Presented results might bring scientific background for future research projects expanding knowledge on this interesting aspect of tryptophan metabolism. 

We are grateful for this comment and fully agree with the reviewer. 

Taken together I am going to conditionally consider changes in the manuscript to be sufficient in authors' favour.